# An integrative study of five biological clocks in somatic and mental health

Rick Jansen[1†]*, Laura KM Han[1†], Josine E Verhoeven[1], Karolina A Aberg[2], Edwin CGJ van den Oord[2], Yuri Milaneschi[1], Brenda WJH Penninx[1]

[1]Department of Psychiatry, Amsterdam UMC, Vrije Universiteit Amsterdam, Amsterdam Public Health Research Institute and Amsterdam Neuroscience, Amsterdam, Netherlands; [2]Center for Biomarker Research and Precision Medicine, Virginia Commonwealth University, Richmond, United States

**Abstract** Biological clocks have been developed at different molecular levels and were found to be more advanced in the presence of somatic illness and mental disorders. However, it is unclear whether different biological clocks reflect similar aging processes and determinants. In ~3000 subjects, we examined whether five biological clocks (telomere length, epigenetic, transcriptomic, proteomic, and metabolomic clocks) were interrelated and associated to somatic and mental health determinants. Correlations between biological aging indicators were small (all $r < 0.2$), indicating little overlap. The most consistent associations of advanced biological aging were found for male sex, higher body mass index (BMI), metabolic syndrome, smoking, and depression. As compared to the individual clocks, a composite index of all five clocks showed most pronounced associations with health determinants. The large effect sizes of the composite index and the low correlation between biological aging indicators suggest that one's biological age is best reflected by combining aging measures from multiple cellular levels.

*For correspondence: ri.jansen@ggzingeest.nl

†These authors contributed equally to this work

## Introduction

Aging can be conceptualized in different ways. While chronological age is measured by date of birth, biological age reflects the relative aging of an individual's physiological condition. Biological aging can be estimated by various cellular indices (*López-Otín et al., 2013*). Commonly used indices are based on telomere length, DNA methylation patterns (epigenetic age), variation in transcription (transcriptomic age) as well as alterations in the metabolome (metabolomic age) and in the proteome (proteomic age) (see *Han et al., 2019*; *Xia et al., 2017* and *Jylhävä et al., 2017* for recent reviews). Biological aging is defined as the residuals of regressing predicted biological age on chronological age: a positive value indicates that the biological age is larger than the chronological age. Advanced biological aging (i.e. an increased biological clock) has been associated to poor somatic health, including the onset of aging-related somatic diseases such as cardiovascular disease, diabetes, and cognitive decline (*Xia et al., 2017*). Advanced biological aging has also been correlated to mental health: childhood trauma (*Li et al., 2017*), psychological stress, and psychiatric disorders (*Darrow et al., 2016*; *Han et al., 2018*). Specifically, telomere length has been most extensively researched and was found to be shorter in various somatic conditions (*Jin et al., 2018*), all-cause mortality (*Mons et al., 2017*; *Wang et al., 2018*) and a range of psychiatric disorders (*Lindqvist et al., 2015*). Advanced epigenetic aging has also been linked to worse somatic health, mortality (*Marioni et al., 2015*), depressive disorder (*Han et al., 2018*; *Whalley, 2017*), and post-traumatic stress disorder (*Wolf et al., 2018*), although some studies have found associations with the opposite direction of effect (*Verhoeven et al., 2018*; *Boks et al., 2015*). Advanced transcriptomic aging was found in those with higher blood pressure, cholesterol levels, fasting glucose, and

body mass index (BMI) (*Peters et al., 2015*). Advanced metabolomic aging increases risk on future cardiovascular disease, mortality, and functionality (*Akker et al., 2019*).

While all biological clocks aim to measure the biological aging process, there is limited evidence for cross-correlations among different clocks. Belsky and colleagues (*Belsky et al., 2017*) recently showed low agreement between eleven quantifications of biological aging including telomere length, epigenetic aging, and biomarker-composites. In contrast, *Hastings et al., 2019* showed relatively strong correlations (r > 0.50) between three physiological composite biological clocks (i.e. homeostatic dysregulation, Klemer and Doubal's method and Levine's method), but not with telomere length. Other studies showed that telomere length was not correlated with epigenetic aging (*Han et al., 2018*; *Marioni et al., 2018*), although cell type composition adjustments revealed a modest association (*Chen et al., 2017*). Further, both Hannum and Horvath epigenetic clocks (*Hannum et al., 2013*; *Horvath et al., 2012*) showed modest correlations to a transcriptomic clock.

Most previous studies, however, have separately considered the relation between a single biological clock and different somatic and mental health conditions. To date, extensive integrated analyses across multiple cellular and molecular aging markers in one study are lacking and it remains unknown to what extent different biological clocks are similarly associated to different health determinants. In addition, most studies did not examine health in its full range and, consequently, whether both somatic and mental health are associated with biological aging remains elusive. As it is unlikely that a single biological clock can fully capture the complexity of the aging process (*Cole et al., 2019*), a composite index, that integrates the different biological clocks and thereby aging at several molecular levels, may reveal the strongest health impact. Therefore, there is an additional need to integrate different biological clocks and test whether such a 'composite clock' outperforms single biological blocks in its association with health determinants.

To develop a better understanding of the mechanisms underlying biological aging, this study aimed to examine (1) the intercorrelations between biological aging indicators based on different molecular levels ranging from DNA to metabolites, namely telomere length, epigenetic, transcriptomic, proteomic and metabolomic clocks; (2) the relationships between different biological aging indicators with both somatic and mental health determinants; and (3) whether a composite biological clock outperforms single biological aging indicators in its association with health. For the five biological aging indicators and the composite clock, associations were computed with a wide panel of lifestyle (e.g. alcohol use, physical activity, smoking), somatic health (functional indicators, BMI, metabolic syndrome, chronic diseases) and mental health (childhood trauma, depression status) determinants.

## Results

### Sample characteristics

To create indicators for biological aging we used whole blood derived measurements from the Netherlands Study of Depression and Anxiety (NESDA) baseline assessment: telomere length (*N* = 2936), epigenetics (DNA methylation, *N* = 1130, MBD-seq, 28M CpGs), gene expression (*N* = 1990, Affymetrix U219 micro arrays, >20K genes), proteomics (*N* = 1837, Myriad RBM DiscoveryMAP 250+, 171 proteins) and metabolites (*N* = 2910, Nightingale Health platform, 231 metabolites), with 653 overlapping samples (see *Table 1* for sample characteristics). Each subsample included around 66% female, with mean age of around 42 years.

### Computing biological clocks

The methods for creating the biological clocks are described in detail in the Materials and methods section. In brief, for each of the four omics measures (epigenetic, transcriptomic, metabolomic and proteomic) we estimated biological age using ridge regression and cross validation (see *Figure 1* for study design). As telomere length values usually decline with increasing chronological age, this indicator was multiplied by −1 to be able to compare directions of effects consistent with the other biological clocks. Correlations between chronological age and predicted biological age were 0.30 for telomere length, 0.95 for epigenetic age, 0.72 for transcriptomic age, 0.85 for proteomic age, and 0.70 for metabolomic age (*Figure 1*). For each omics-based biological clock, biological aging is defined as the residuals of regressing predicted biological age on chronological age: a positive value

**Table 1.** Sample description.

| | | Telomere Length | Epigenetic Aging | Transcriptomic Aging | Proteomic Aging | Metabolomic Aging | Composite Index |
|---|---|---|---|---|---|---|---|
| | # Subjects | 2936 | 1130 | 1990 | 1837 | 2910 | 653 |
| Demographic | Sex (%female) | 66.00 | 65.00 | 67.00 | 67.00 | 66.00 | 66.00 |
| | Education years (mean) | 12.15 | 11.93 | 12.07 | 12.07 | 12.15 | 11.71 |
| | Age (mean) | 41.81 | 41.53 | 38.71 | 41.37 | 41.94 | 41.23 |
| Lifestyle | Alcohol use (units per week, mean) | 6.24 | 6.54 | 6.38 | 6.39 | 6.29 | 6.48 |
| | Smoking (pack years, mean) | 11.00 | 11.43 | 11.84 | 10.37 | 11.12 | 10.90 |
| | Physical activity (MET minutes per week, mean) | 3679.72 | 3638.54 | 3729.20 | 3741.00 | 3668.13 | 3525.05 |
| Somatic Health | BMI (mean) | 25.60 | 25.67 | 25.68 | 25.66 | 25.60 | 25.82 |
| | Physical disability (score, mean) | 24.40 | 29.45 | 26.00 | 23.22 | 24.45 | 30.27 |
| | Lung capacity (PEF in liter/minute, mean) | 477.74 | 479.75 | 478.42 | 477.19 | 477.23 | 475.23 |
| | Hand grip strength (kg, mean) | 37.06 | 37.77 | 37.08 | 37.46 | 37.05 | 37.74 |
| | Cardiometabolic disease (%cases) | 18 | 18 | 18 | 18 | 18 | 17 |
| | Respiratory disease (%cases) | 9 | 9 | 9 | 9 | 9 | 10 |
| | Musculoskeletal disease (%cases) | 10 | 10 | 10 | 9 | 10 | 9 |
| | Digestive disease (%cases) | 9 | 9 | 9 | 8 | 9 | 8 |
| | Neurological disease (%cases) | 3 | 2 | 3 | 3 | 3 | 2 |
| | Endocrine disease (%cases) | 3 | 3 | 3 | 3 | 3 | 4 |
| | Cancer (%cases) | 7 | 8 | 7 | 7 | 7 | 8 |
| | Metabolic syndrome (# components, mean) | 1.36 | 1.39 | 1.37 | 1.33 | 1.36 | 1.41 |
| | # Chronic diseases (mean) | 0.61 | 0.62 | 0.62 | 0.58 | 0.61 | 0.63 |
| Mental Health | Current MDD (%cases) | 27 | 72 | 34 | 26 | 27 | 76 |
| | Depression severity (IDS, mean) | 21.46 | 25.80 | 22.91 | 20.96 | 21.48 | 26.67 |
| | Childhood Trauma (score from 0-4, mean) | 0.91 | 0.97 | 1.00 | 0.87 | 0.92 | 1.01 |

means that the biological age is larger than the chronological age. The individual clocks residualized for chronological age are also referred to as biological aging indicators. Correlations between biological aging indicators, corrected for sex, are presented in *Figure 2*. Correlations were significant for 3 out of 10 pairs; proteomic vs metabolomic aging (*r* = 0.19, p=2e-16), transcriptomic vs epigenetic aging (*r* = 0.15, p=3e-06) and transcriptomic vs proteomic aging (*r* = 0.08, p=2e-06).

## Associations between individual biological aging indicators and health determinants

For each of the five biological aging indicators, we computed associations with several demographic (sex, education), lifestyle (physical activity, smoking, alcohol use), somatic health (BMI, hand grip strength, lung function, physical disability, chronic diseases), and mental health (current depression, depression severity, childhood trauma) determinants. Except for proteomic aging, sex was associated with all biological aging indicators: women were biologically younger than men (p=3e-4 for telomere length, p=5e-4 for epigenetic aging, p=4e-11 for transcriptomic aging, p=1e-5 for metabolomic aging). Education was not associated with any biological aging indicator. We controlled for sex by using it as a covariate in all following models (except for in the model where sex was the outcome). *Table 2* and *Figure 3* give an overview of all associations. Correction for multiple testing was done using permutation-based FDR (Materials and methods), resulting in a p-value threshold of 2e-2 for an FDR of 5% for all tests.

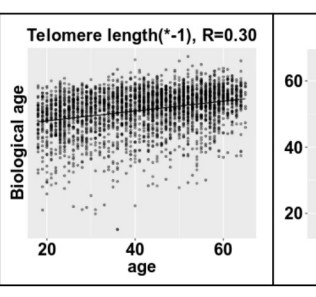
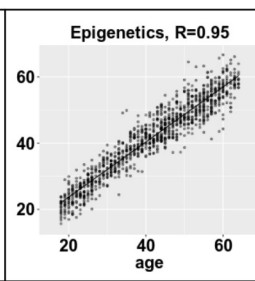
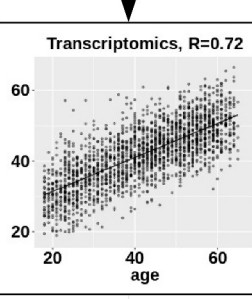
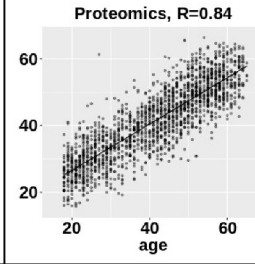
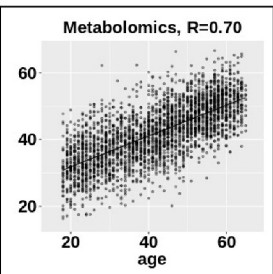

### Telomere

-Leukocyte DNA
-qPCR
-Telomere length
-*N*=2936

### Epigenetics

-Whole blood DNA
-MBD-seq
-28M CpGs
-*N*=1130

### Transcriptomics

-Whole blood RNA
-Micro arrays
-18K genes
-*N*=1990

### Proteomics

-Whole blood serum
-Immunoassay
-171 proteins
-*N*=1837

### Metabolomics

-Whole blood plasma
-Nightingale platform
-231 Metabolites
-*N*=2910

Estimate biological age

Compute biological aging: regress biological age on chronological age.
Compute composite index of 5 biological clocks.
Compute associations with:

### Demographics & lifestyle

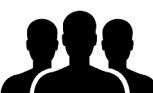

-Sex
-Education
-Physical activity
-BMI
-Smoking
-Alcohol use

### Somatic health

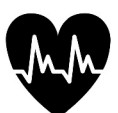

-Hand grip strength
-Lung function
-Physical functioning
-Chronic diseases

### Mental health

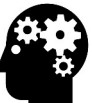

-Current depression
-Depression severity
-Childhood trauma

**Figure 1.** Study design. The upper part of the figure shows the five biological layers. From each of the four omics layers (epigenetic, transcriptomic, proteomic, and metabolomic data), biological age was estimated, and biological age was regressed on age to obtain measures of biological aging. Only telomere length was not age-regressed. The five biological aging indicators were associated with multiple demographic, lifestyle, somatic health and mental health determinants.

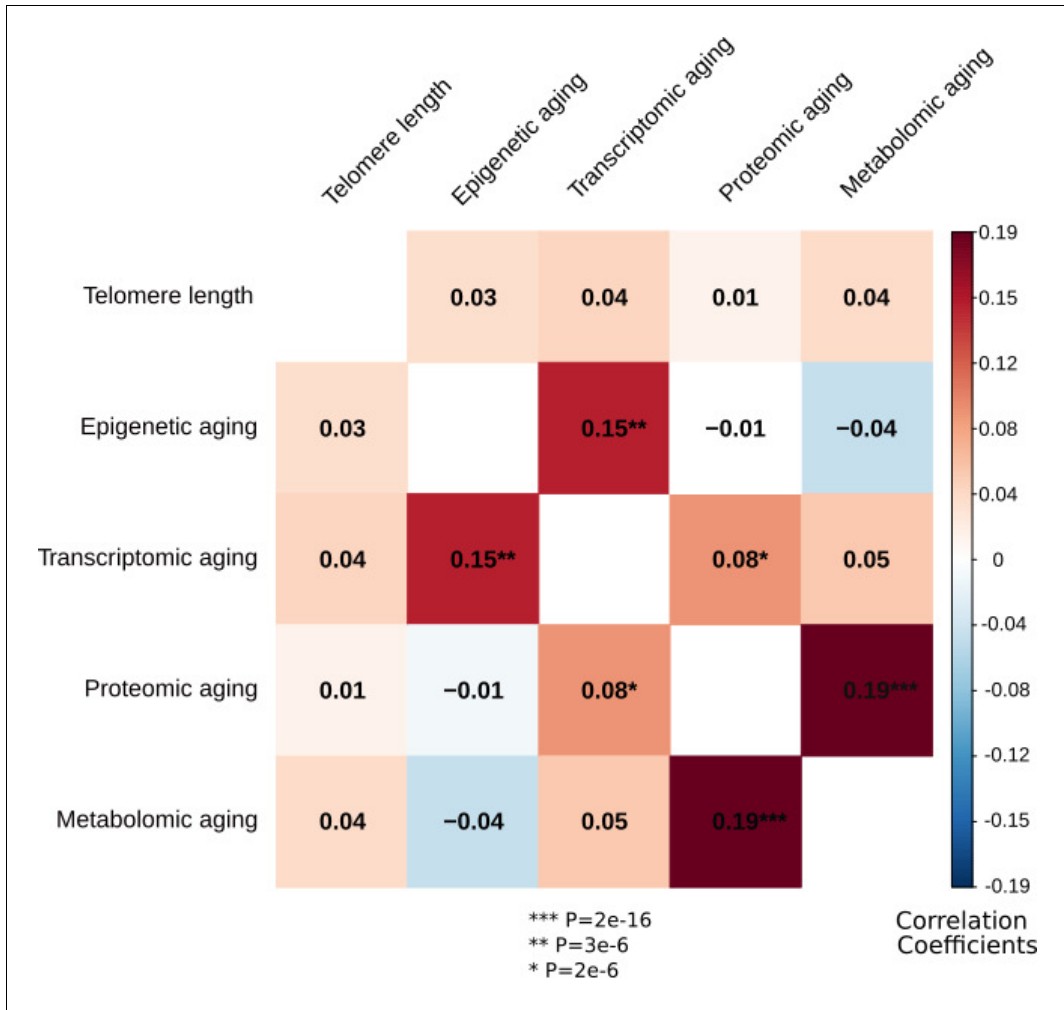

**Figure 2.** Correlations between the biological aging indicators. The heatmap represents Spearman rank correlations between the five biological aging indicators, all corrected for sex. Out of 10 pairs, three are significant: transcriptomic vs epigenetic aging, metabolomic vs proteomic aging and proteomic vs transcriptomic aging. All biological aging indicators were age-regressed, only telomere length was not.

Among the lifestyle determinants, alcohol use was associated with advanced proteomic aging (p=3e-3) and smoking (packs per year) was associated with shorter telomere length (p=3e-3), and advanced transcriptomic (p=2e-2), proteomic (p=1e-5) and metabolomic aging (p=5e-3). Physical activity was not associated with any biological aging indicator.

From the somatic health determinants, high BMI was strongly associated with advanced biological aging of all indicators (p=2e-2 for telomere length, p=4e-3 for epigenetic aging, p=6e-10 for transcriptomic aging, p=1e-7 for proteomic aging, and p=2e-35 for metabolomic aging). Physical disability was associated with advanced epigenetic aging (p=1e-4). Within the domain of chronic diseases, the presence of digestive diseases and endocrine diseases were associated with advanced proteomic aging (p=2e-2 and p=1e-2, respectively). Subjects with cardiometabolic disease showed advanced metabolomic aging (p=4e-3) and subjects with digestive disease exhibited advanced transcriptomic aging (p=1e-2). Those with metabolic syndrome showed advanced biological aging across four indicators (p=6e-4 for telomere length, p=1e-8 for transcriptomic aging, p=5e-9 for proteomic aging, p=5e-29 for metabolomic aging).

The presence of current depression and depression severity were associated advanced epigenetic (p=2e-3 and p=9e-5) and proteomic aging (p=8e-3 and p=6e-3, respectively). Current depression was also associated with advanced transcriptomic aging (p=2e-2) and those with childhood trauma showed advanced epigenetic aging (p=8e-5). To verify if the results were confounded by medication

**Table 2.** Associations between five biological aging indicators and multiple health determinants.

For each biological aging indicator, linear models were fit with the health determinant as predictor, while controlling for sex. Beta's and p-values from these models are presented here. In the 653 samples with all five data layers available, a composite index was constructed which was significantly associated with more variables than any of the five individual biological aging indicators. All biological aging indicators were age-regressed, only telomere length was not. Telomere length models were corrected for age instead. * Beta for telomere length was multiplied by −1 to compare with other biological aging indicators. All measures are coded such that higher values indicate advanced biological aging. Bold indicates FDR < 5%.

| | | Telomere Length (N=2936) | | Epigenetic clock (N=1130) | | Transcriptomic Clock (N=1990) | | Proteomic Clock (N=1837) | | Metabolomic Clock (N=2910) | | Composite Index (sum) (N=653) | | Composite Index (PC1) (N=653) | |
|---|---|---|---|---|---|---|---|---|---|---|---|---|---|---|---|
| | | Beta* | P | Beta | P | Beta | P | Beta | P | Beta | P | Beta | P | Beta | P |
| Demographic | Sex (male/female) | -0.06 | **2.89E-04** | -0.10 | **4.65E-04** | -0.15 | **3.64E-11** | -0.03 | 1.46E-01 | -0.08 | **1.25E-05** | -0.18 | **2.33E-06** | -0.11 | **3.59E-03** |
| | Education (# years) | -0.03 | 1.12E-01 | -0.02 | 5.21E-01 | -0.01 | 6.37E-01 | -0.05 | 3.43E-02 | -0.03 | 8.22E-02 | -0.04 | 3.11E-01 | -0.05 | 2.27E-01 |
| Lifestyle | Alcohol use (units per week) | 0.03 | 1.05E-01 | -0.05 | 1.40E-01 | 0.00 | 9.21E-01 | 0.07 | **2.89E-03** | 0.04 | 4.57E-02 | 0.07 | 6.05E-02 | 0.09 | **1.50E-02** |
| | Smoking (pack years) | 0.06 | **3.11E-03** | 0.02 | 6.22E-01 | 0.05 | **1.55E-02** | 0.10 | **1.33E-05** | 0.05 | **5.09E-03** | 0.10 | **1.15E-02** | 0.12 | **2.85E-03** |
| | Physical activity | 0.02 | 2.75E-01 | -0.06 | 3.88E-02 | -0.04 | 6.42E-02 | 0.03 | 1.51E-01 | 0.01 | 5.18E-01 | -0.04 | 3.62E-01 | 0.01 | 7.38E-01 |
| Somatic Health | BMI | 0.04 | **1.80E-02** | 0.09 | **3.94E-03** | 0.14 | **6.02E-10** | 0.12 | **9.82E-08** | 0.23 | **2.07E-35** | 0.24 | **2.32E-10** | 0.22 | **2.18E-09** |
| | Physical disability | 0.03 | 9.11E-02 | 0.11 | **1.41E-04** | 0.04 | 8.61E-02 | 0.04 | 7.42E-02 | -0.01 | 4.24E-01 | 0.10 | **7.38E-03** | 0.03 | 4.01E-01 |
| | Lung capacity | 0.02 | 4.19E-01 | 0.03 | 4.65E-01 | 0.04 | 2.13E-01 | -0.04 | 1.51E-01 | 0.03 | 2.37E-01 | 0.03 | 5.34E-01 | -0.02 | 6.57E-01 |
| | Hand grip strength | -0.02 | 3.33E-01 | -0.06 | 1.71E-01 | 0.03 | 3.52E-01 | 0.01 | 7.30E-01 | 0.03 | 2.24E-01 | -0.03 | 6.14E-01 | 0.03 | 6.20E-01 |
| | Cardiometabolic disease (no/yes) | 0.02 | 3.37E-01 | 0.04 | 1.56E-01 | 0.03 | 1.44E-01 | 0.03 | 1.35E-01 | 0.05 | **3.94E-03** | 0.10 | **1.37E-02** | 0.08 | 3.19E-02 |
| | Respiratory disease (no/yes) | -0.02 | 2.12E-01 | -0.01 | 6.34E-01 | 0.02 | 2.85E-01 | 0.03 | 1.27E-01 | 0.01 | 4.67E-01 | -0.03 | 4.70E-01 | 0.01 | 7.17E-01 |
| | Musculoskeletal disease (no/yes) | 0.00 | 8.11E-01 | -0.01 | 7.37E-01 | 0.04 | 1.04E-01 | 0.02 | 4.36E-01 | 0.02 | 2.23E-01 | 0.09 | 2.27E-02 | 0.11 | **4.96E-03** |
| | Digestive disease (no/yes) | 0.03 | 5.77E-02 | -0.02 | 5.71E-01 | 0.06 | **9.76E-03** | 0.06 | **1.21E-02** | 0.02 | 2.81E-01 | 0.05 | 2.01E-01 | 0.04 | 2.86E-01 |
| | Neurological disease (no/yes) | -0.02 | 2.58E-01 | 0.02 | 5.60E-01 | 0.01 | 5.44E-01 | 0.02 | 2.84E-01 | 0.02 | 1.93E-01 | -0.04 | 2.64E-01 | -0.02 | 5.09E-01 |
| | Endocrine disease (no/yes) | -0.01 | 4.45E-01 | 0.01 | 8.13E-01 | -0.01 | 5.75E-01 | 0.06 | **1.03E-02** | 0.03 | 1.23E-01 | 0.06 | 1.18E-01 | 0.09 | **1.64E-02** |
| | Cancer (no/yes) | 0.00 | 9.66E-01 | 0.02 | 5.65E-01 | 0.02 | 4.88E-01 | 0.03 | 1.81E-01 | 0.02 | 2.01E-01 | 0.08 | 3.22E-02 | 0.07 | 5.00E-02 |
| | Metabolic syndrome (# components) | 0.06 | **6.35E-04** | 0.04 | 1.46E-01 | 0.13 | **9.98E-09** | 0.13 | **5.34E-09** | 0.21 | **4.53E-29** | 0.28 | **9.10E-13** | 0.26 | **6.41E-12** |
| | # Chronic diseases | 0.00 | 7.99E-01 | 0.03 | 3.63E-01 | 0.05 | 3.20E-02 | 0.09 | **1.24E-04** | 0.03 | 1.39E-01 | 0.06 | 1.26E-01 | 0.07 | 8.43E-01 |
| Mental Health | Current MDD (no/yes) | 0.03 | 1.59E-01 | 0.09 | **1.99E-03** | 0.07 | **1.68E-02** | 0.08 | **7.62E-03** | -0.03 | 1.61E-01 | 0.11 | **6.05E-03** | -0.12 | 2.29E-01 |
| | Depression severity | 0.04 | 2.40E-02 | 0.12 | **8.67E-05** | 0.03 | 2.76E-01 | 0.07 | **5.99E-03** | -0.02 | 3.74E-01 | 0.13 | **7.61E-04** | 0.05 | 1.87E-01 |
| | Childhood Trauma | 0.01 | 4.54E-01 | 0.12 | **7.99E-05** | 0.03 | 2.06E-01 | 0.04 | 8.96E-02 | 0.04 | 2.46E-02 | 0.09 | **1.96E-02** | 0.07 | 7.19E-02 |

* Beta for telomere length was multiplied by -1 to compare with other biological clocks.
Bold indicates FDR<5%.

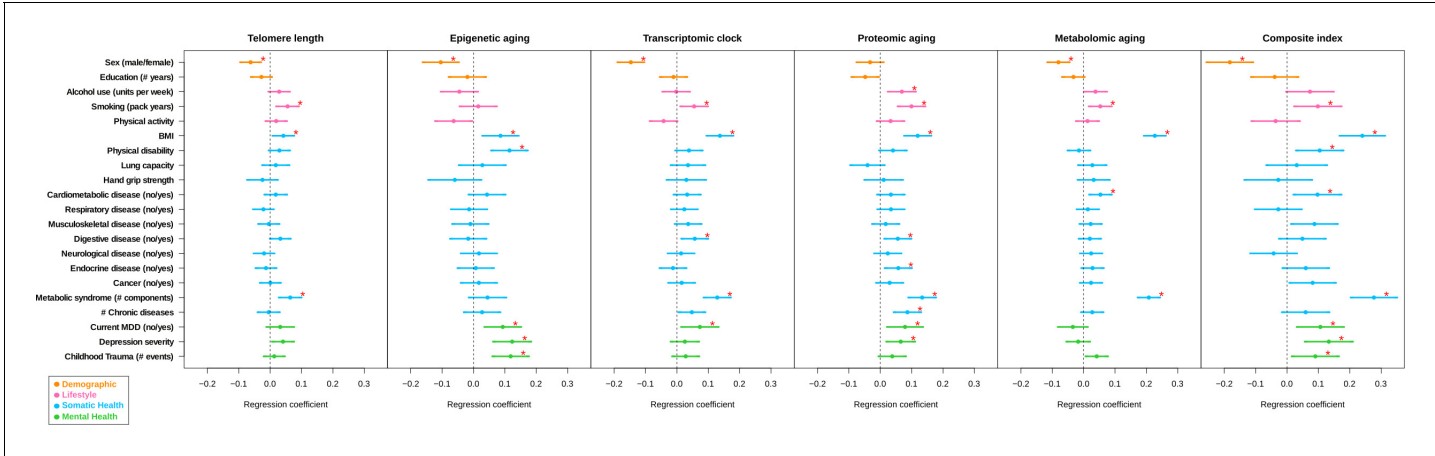

**Figure 3.** Forest plot of associations between biological aging and health determinants. For each of the associations between biological aging indicators and health determinants, the standardized beta and standard deviation derived from linear models were plotted. The significant associations (p<2e-2, FDR < 5%) are shown with red stars. The composite index, which is the scaled sum of the five biological aging indicators, clearly shows most associations and often largest effect sizes. Biological aging was used as outcome in the linear models. Beta for telomere length was multiplied by −1 to compare with other biological clocks. Red stars indicate FDR < 5%. All biological aging indicators were age-regressed, only telomere length was not.

use, we computed associations between antidepressant medication (SSRIs, TCAs, or other antidepressants), metabolic-syndrome-related medication ('metabolic medication': anti-diabetic, fibrates, or anti-hypertensives) and biological aging (*Supplementary file 1*). After FDR correction, we found that metabolomic aging was associated with the increased use of metabolic medication (*Beta = 0.153, p*=2.35e-3), and antidepressant use with proteomic (*Beta = 0.208, p*=7.16e-5) and transcriptomic aging (*Beta = 0.129, p*=8.1e-3). The design of the current observational study cannot conclusively prove whether this is a direct medication effect or confounding by indication.

## Association between biological aging indicators and mortality in longitudinal analysis

We conducted post-hoc analyses on the relationship between the biological aging indicators and subsequent outcomes after 6 years of follow-up duration. Mortality data and self-reported somatic disease onset (in the categories cardiometabolic, respiratory, musculoskeletal, digestive, neurological, and endocrine diseases, and cancer) was gathered at each measurement wave. There were no significant associations between chronic disease onset or mortality and baseline biological aging, likely due to the low numbers of mortality and disease onset (*Supplementary file 3*).

## Associations between the biological aging composite index and all health determinants

The composite index was computed as the sum of the five scaled biological aging indicators in the 653 samples with data of all five biological levels. Correlations between the five biological aging indicators and the composite index were between 0.43 and 0.51. We found more and stronger associations for the composite index than for any of the individual biological aging indicators: including sex (p=2e-6), BMI (p=2e-10), smoking (p=2e-2), metabolic syndrome (p=9e-13), current MDD (p=6e-3), depression severity (p=7e-4), and childhood trauma (p=2e-2). As an alternative approach, Principal Component Analysis (PCA) was used to compute an alternative composite index. We used the first principle component (PC) of this analysis, which was a weighted sum of the biological aging indicators (for telomere length the weight (w) = 0.042, epigenetic aging w = 0.094, transcriptomic aging w = 0.220, proteomic aging w = 0.707, metabolomic aging w = 0.664), reflecting the highest correlations between the biological aging indicators, which is between metabolomic and proteomic aging. Compared to the composite index that was based on the sum and thus gives equal weight to all five biological aging indicators, the PC-based index had less significant associations with sex, smoking, BMI, and metabolic syndrome. The PC-based index was not significantly associated with physical disability, or mental health outcomes, as opposed to the summed index. The five PC's each explain

more than 15% of variance (the first 2 PC's more than 25% each), indicating the multidimensionality and non-redundancy of the five biological aging indicators.

To allow for direct effect size comparisons between the composite (summed) index and the individual aging indicators, we compared the findings for the composite index to those of each individual biological aging indicator with the same subsample. In this analysis, p-values and effect sizes were often more pronounced for the composite index (*Figure 4*, *Supplementary file 2*). For example, sex, BMI, metabolic syndrome and current MDD, were significantly associated with the composite index, but the betas for the composite index were larger than the betas from any individual indicator. For the other five variables significantly associated with the composite index (smoking, physical disability, cardiometabolic disease, depression severity, and childhood trauma) the betas for the composite index were larger than four out of five betas from the individual biological aging indicators.

## Discussion

In this study, we examined five biological clocks based on telomere length and four omics levels from a large, clinically well-characterized cohort. We demonstrated significant intercorrelations between three pairs of biological aging indicators, illustrating the complex and multifactorial processes of biological aging. Furthermore, we observed both overlapping and unique associations between the individual biological aging indicators and different lifestyle, somatic and mental health determinants. Separate linear regressions showed that male sex, high BMI, smoking, and metabolic syndrome were consistently associated with more advanced levels of biological aging across at least four of the biological clocks. Strikingly, depression was associated to more advanced levels of epigenetic, transcriptomic and proteomic aging, signifying that both somatic and mental health are associated with advanced biological aging. Finally, by integrating a composite index of all biological aging indicators we were able to obtain larger effect sizes with for example physical disability and childhood trauma exposure, underscoring the broad impact of determinants on cumulative multi-system biological aging.

The range of correlations among the biological aging indicators considered in this study indicates that the correlates of chronological age in different molecular layers were not strongly correlated, suggesting that biological aging may be differently manifested at certain cellular levels. Consistent with prior studies, we showed weak correlations between different biological aging indicators (*Li et al., 2020*) and we confirm the absent relationship between telomere length and epigenetic aging (*Marioni et al., 2018*; *Belsky et al., 2017*; *Breitling et al., 2016*), but also show lack of associations with transcriptomic, proteomic or metabolomic aging. However, we do confirm an earlier finding showing a significant but modest correlation between epigenetic and transcriptomic aging (*Peters et al., 2015*). The correlation between metabolomic and proteomic aging may partly be explained by the fact that both data were obtained from platforms that were aimed at probing central inflammation lipid processes, rather than the full proteome or metabolome. Nevertheless, we can infer that only some biological aging indicators show overlap, while most of them seem to be tracking distinctive parts of the aging process, even if they are associated with the same somatic or health determinants.

Our study showed that several of the determinants considered exhibited consistent associations with different biological aging indicators. First, male sex was associated with shorter telomere length and advanced epigenetic, transcriptomic, and metabolomic aging, in line with a large body of literature that shows advanced biological aging and earlier mortality in males compared to females (*Austad and Fischer, 2016*). Second, high BMI was consistently related to all biological aging indicators, showing that the more overweight or obese, the higher the biological age (*Gielen et al., 2018*), also after controlling for sex. Earlier studies showed similar associations between high BMI and shorter telomere length (*Gielen et al., 2018*), and older epigenetic (*Horvath and Raj, 2018*) and transcriptomic aging signatures (*Peters et al., 2015*). Third, our analyses showed similarly consistent associations between the prevalence of metabolic syndrome and advanced levels of aging. Further, all but epigenetic aging was advanced with respect to cigarette smoking.

Major depressive disorder (MDD) status was consistently related to advanced aging in three (epigenetic, transcriptomic, proteomic) out of the five biological aging indicators. In contrast, a recent study ($N > 1000$) in young adults (20–39 years) did not show associations between mental health (as

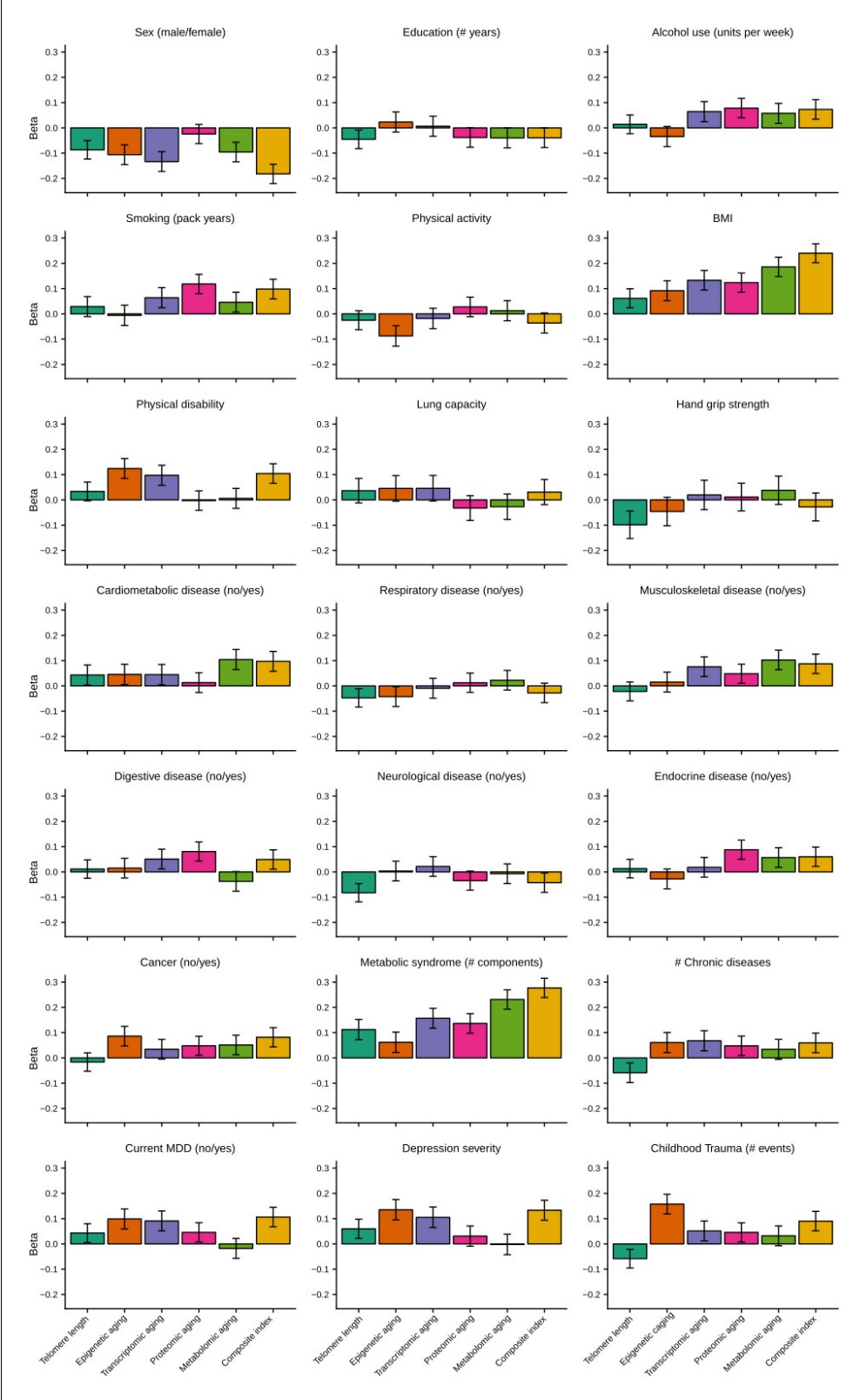

**Figure 4.** Barplots of betas from associations between biological aging and health determinants. For each of the associations between biological aging and health determinants, the standardized beta and standard deviation derived from linear models were plotted. Only samples that had data for all five biological clocks (*N* = 653) were used. All biological aging indicators were age-regressed, only telomere length was not.

measured by the CIDI) and biological aging (indicated by telomere length, homeostatic dysregulation, Klemer and Doubal's method and Levine's method) (*Hastings et al., 2019*), but it seems possible that this sample was too young to fully develop aging-related manifestations of mental health problems, or lacked age variation. It is likely that our data (obtained from participants 18–64 years) may have been more sensitive in picking up associations with mental health due to increased variation in both chronological age (i.e. inclusion of older persons), as well as symptom severity. To further examine whether the results were consistent across participants with and without depressive psychopathology, we repeated all models in post-hoc analyses and added an interaction term between current depression status and health determinants. There was an overall consistent pattern of non-significant interaction terms for most health determinants and biological aging, although only higher BMI was significantly associated to advanced epigenetic aging in the psychopathology group. However, taken together, the results suggest that findings are not different in persons with and without mental disorders. We observed some significant associations between biological aging and medication use. The design of the current observational study cannot conclusively prove whether this is a direct medication effect or confounding by indication: the patient group using antidepressant medication is also the group that is more chronically and severely depressed. This is similar for the metabolic syndrome related medication. Future studies using randomized clinical trial designs are needed to investigate the mechanism of action of direct pharmacological effects of medication on biological aging.

Furthermore, we computed a composite index by summing up the five biological aging indicators studied here. In other words, this integrative metric contains cumulative independent signal from the individual markers and dependent shared signal – with possible reduced noise due to the summation – between them. Given that this composite index demonstrates larger effect sizes for BMI, sex, smoking, depression severity, and metabolic syndrome than the individual aging indicators, it is suggested that being biologically old at multiple cellular levels has a cumulative multi-systemic effect. When integrated, the composite index reveals stronger (i.e. greater cumulative betas for the composite index than individual clocks) converging associations with sex, BMI, metabolic syndrome and current MDD. This provides further support for the hypothesis that not one biological clock sufficiently captures the biological aging process and that not all clocks are under the control of one unitary aging process. There is abundant room for further progress in determining whether biological aging can be modified by intervening on these determinants.

Nonetheless, the question remains which biological mechanism could plausibly link the current quantification of biological aging and its lifestyle, somatic, and mental health determinants. Part of this answer requires discussion on the features used to build the different clocks: the proteomic and metabolomic clocks mostly measure inflammatory or metabolic factors, two highly integrated processes in aging and aging-related diseases (*Frasca et al., 2017*). Previous studies suggest immune-mediated mechanisms (specifically inflammatory signaling) connecting metabolic syndrome (*Révész et al., 2015*), mental health disorders (*Wohleb et al., 2016*), and aging (*Révész et al., 2018*). Moreover, MDD is a condition in which inflammation, obesity, and premature or advanced aging co-occur and converge. It might therefore be speculated that immunity and 'inflammaging' (*Franceschi et al., 2018*) may tie together the currently observed associations.

This study did not include existing biological clocks. While the application of established algorithms would increase generalizability of our findings, there are several reasons why it was not optimal to implement previously published algorithms in the NESDA data. First and foremost, generated omics data are platform-dependent and the existing epigenetic (*Horvath, 2013*) and gene expression (*Peters et al., 2015*) clocks rely on arrays with different coverage of probes as was used in NESDA, that also target different parts of genes. Second, a subsample of NESDA was part of the previously published metabolomic clock (*Akker et al., 2019*), thus application of this model to the current dataset would result in overfitting. The current proteomic platform has not been used before to train a biological clock. Moreover, there is currently no validated gold standard for calculating transcriptomic, proteomic, or metabolomic clocks. Importantly, in spite of these limitations, we have followed an alternative but consistent methodological approach for training our omics-based biological clocks, leveraging the advantage to compare, combine, and integrate these clocks within the same population. However, we emphasize the need for epidemiological replication of these determinants in other datasets (e.g. those including different ethnicities) and we recognize

that data harmonization and pooling are important strategies on the scientific research agenda that may overcome this limitation in the future.

Since no previously published algorithms were used, we trained our own clocks using ridge regression with cross-validation. This approach relies on the assumption that the determined cross-sectional correlation between the omics patterns and chronological age arise mainly as a consequence of biological aging, and is independent from potential secular trends (*Nelson et al., 2020*; *Belsky, 2015*; *Belsky et al., 2020*). As common to cross-sectional studies, it is, however, impossible to completely rule out potential cohort effects or uncontrolled individual differences and results should be interpreted in light of this limitation. Future longitudinal studies are needed to identify patterns of biological changes that go beyond their ability to predict age at the time of sampling. While the current study only used chronological age as criterion endpoint, it is important to mention that other epigenetic clocks exist that are trained to predict other potential criteria such as phenotypic markers of age (DNAm PhenoAge) (*Levine et al., 2018*) or a composite biomarker that was derived from DNAm surrogates and smoking in pack-years (GrimAge) (*Hillary et al., 2019*). Such clocks were developed to lead to improved predictions of risk of mortality.

More research is needed to elucidate whether: (1) physiological disturbances, such as loss of inflammatory control associated with somatic and psychopathology, accelerate biological aging over time, (2) advanced biological aging precedes and constitutes a vulnerability factor that causes somatic and psychopathology, or (3) somatic and psychopathology and biological aging processes are not causally linked, but share underlying etiological roots (e.g. shared genetic risks or environmental factors) (*Han et al., 2019*). Yet, it could conceivably be hypothesized that dysregulation of immunoinflammatory control may be related to metabolic outcomes, aging, and depression (*Diniz and Vieira, 2018*), providing scope as to why some of these determinants converge across different platforms and multiple biological levels.

Here, we used a large cohort that was well-characterized in terms of demographics, lifestyle, and both somatic and mental health assessments, to study and integrate five biological clocks across multiple levels of analysis. This is particularly important as we show that the determinants of biological aging encompass several different domains. Moreover, our sample was adequately powered to detect statistically significant associations, limiting the possibility for chance findings and increasing probability for identifying robust biological age determinants. On the other hand, an obvious limitation is the cross-sectional nature of this study that prevents us from drawing any conclusions on whether the determinants accelerate the aging trajectory over time, the other way around, or whether 'third' variables effect this association.

Another aspect that limits the interpretability of our findings in the context of increased risk of developing aging-related diseases and mortality was the relatively young age of the current sample. To illustrate, we were unable to predict future incidence of chronic disease or mortality from baseline biological aging, likely due to the low numbers of mortality and disease onset (*Supplementary file 3*), for example the number of deceased cases ranged from 64 (TL) to 27 (proteomic clock). Previous studies that have associated biological aging with mortality risk commonly include aging cohorts (Danish longitudinal twin study with mean age of 86.1 years; Framingham Offspring Study with mean age 61.0 years; Swedish population cohort SATSA with mean age 63.6 years; German population cohort ESTHER with mean age 62.5 years; Lothian Birth Cohorts with mean age >69.5 years; Normative Aging Study with mean age 71.7 years) (*Marioni et al., 2018*; *Li et al., 2020*; *Christiansen et al., 2016*; *Perna et al., 2016*; *Murabito et al., 2018*; *Chen et al., 2016*). Before definitively interpreting a 'clock' as a measure of biological aging, further independent studies are needed to establish that the clock changes with advancing age and forecasts disease, disability and mortality.

## Conclusions

In conclusion, this study examined the overlap between five biological aging indicators and their shared and unique associations with somatic and mental health. Our findings indicate that they largely track distinct, but also partially overlapping aspects of this aging process. Further, we demonstrated that male sex, smoking, higher BMI and metabolic syndrome were consistently related to advanced aging at multiple biological levels. Remarkably, our study also converges evidence of depression and childhood trauma associations across multiple platforms, cellular levels, and sample sizes, highlighting the important link between mental health and biological aging. Taken together,

our findings contribute to the understanding and identification of biological age determinants, important to the development of end points for clinical and epidemiological research.

## Materials and methods

### Study design and participants

Data used were from the Netherlands Study of Depression and Anxiety (NESDA), an ongoing longitudinal cohort study examining course and consequences of depressive and anxiety disorders. The NESDA sample consists of 2981 persons between 18 and 65 years including persons with a current or remitted diagnosis of a depressive and/or anxiety disorder (74%) and healthy controls (26%). Individuals were recruited from mental health care settings, general practitioners, and the general population in the period from September 2004 to February 2007. Persons with insufficient command of the Dutch language or a primary clinical diagnosis of other severe mental disorders, such as severe substance use disorder or a psychotic disorder were excluded. Participants were assessed during a 4 hr clinical visit, consisting of the collection of all somatic and mental health determinants in the current study, as well as a fasting blood draw. All omics data was obtained from the same blood sample, drawn at the same time point as the health determinant examination during the face-to-face visit. The study was approved by the Ethical Review Boards of participating centers, and all participants signed informed consent. More than 94% of the NESDA participants were from North European origin. The population and methods of the NESDA study have been described in more detail elsewhere (*NESDA Research Consortium et al., 2008*).

Data to derive different biological clocks was available for different subsamples and all based on the same fasting blood draw from participants in the morning between 8:30 and 9:30 after which samples were stored in a −80°C freezer or – for RNA - transferred into PAXgene tubes (Qiagen, Valencia, California, USA) and stored at −20°C. To create biological clocks, we used telomere length (N = 2936), DNA methylation (N = 1130, MBD-seq, 28M CpGs), gene expression (N = 1990, Affymetrix U219 micro arrays, >20K genes), proteins (N = 1837, Myriad RBM DiscoveryMAP 250+, 171 proteins) and metabolites (N = 2910, Nigthingale platform, 231 metabolites), see *Table 1* and details in the following sections.

### Biological clock assessments

#### Telomere length

Leukocyte telomere length was determined at the laboratory of Telomere Diagnostics, Inc (Menlo Park, CA, USA), using quantitative polymerase chain reaction (qPCR), adapted from the published original method by *Cawthon, 2002*. Telomere sequence copy number in each patient's sample (T) was compared to a single-copy gene copy number (S), relative to a reference sample. The resulting T/S ratio is proportional to mean leukocyte telomere length. The detailed method is described elsewhere (*Verhoeven et al., 2014*). The reliability of the assay was adequate: eight included quality control DNA samples on each PCR run illustrated a small intra-assay coefficient of variation (CV = 5.1%), and inter-assay CV was also sufficiently low (CV = 4.6%).

#### DNA methylation (epigenetic clock)

To assay the methylation levels of the approximately 28 million common CpG sites in the human genome, we used an optimized protocol for MBD-seq (*Han et al., 2018*; *Aberg et al., 2020*). With this method, genomic DNA is first fragmented and the methylated fragments are then bound to the MBD2 protein that has high affinity for methylated DNA. The non-methylated fraction is washed away and only the methylation-enriched fraction is sequenced. This optimized protocol assesses about 94% of the CpGs in the methylome. The sequenced reads were aligned to the reference genome (build hg19/GRCh37) with Bowtie2 (*Langmead and Salzberg, 2012*) using local and gapped alignment. Aligned reads were further processed using the RaMWAS Bioconductor package (*Shabalin et al., 2018*) to perform quality control and calculate methylation scores for each CpG.

## Gene expression (transcriptomic clock)

RNA processing and assaying -done at Rutgers University Cell and DNA repository- have been described previously (*Jansen et al., 2014*; *Jansen et al., 2017*; *Wright et al., 2014*). Samples were hybridized to Affymetrix U219 arrays (Affymetrix, Santa Clara, CA). Array hybridization, washing, staining, and scanning were carried out in an Affymetrix GeneTitan System per the manufacturer's protocol. Gene expression data were required to pass standard Affymetrix QC metrics (Affymetrix expression console) before further analysis. We excluded from further analysis probes that did not map uniquely to the hg19 (Genome Reference Consortium Human Build 37) reference genome sequence, as well as probes targeting a messenger RNA (mRNA) molecule resulting from transcription of a DNA sequence containing a single nucleotide polymorphism (based on the dbSNP137 common database). After this filtering step, data for analysis remained for 423,201 probes, which was summarized into 44,241 probe sets targeting 18,238 genes. Normalized probe set expression values were obtained using Robust Multi-array Average (RMA) normalization as implemented in the Affymetrix Power Tools software (APT, version 1.12.0, Affymetrix). Data for samples that displayed a low average Pearson correlation with the probe set expression values of other samples, and samples with incorrect sex-chromosome expression were removed.

## Proteins (proteomic clock)

As described previously (*Lamers et al., 2016*), a panel of 243 analytes (Myriad RBM DiscoveryMAP 250+) involved in various hormonal, immunological, and metabolic pathways was assessed in serum using multiplexed immunoassays in a Clinical Laboratory Improvement Amendments (CLIA)-certified laboratory (Myriad RBM; Austin, TX, USA). After excluding analytes with more than 30% missing data (mostly due to values outside the ranges of detection), 171 of the 243 analytes remained for analysis (with values below and above detection limits imputed with the detection limit values).

## Metabolites (metabolomic clock)

Metabolite measurements have been described in detail previously (*Akker et al., 2019*; *BBMRI-NL Metabolomics Consortium et al., 2020*). In short, a total of 232 metabolites or metabolite ratios were reliably quantified from Ethylenediaminetetraacetic acid plasma samples using targeted high-throughput proton Nuclear Magnetic Resonance ($^1$H-NMR) metabolomics (Nightingale Health Ltd, Helsinki, Finland) (*Soininen et al., 2015*). Metabolites measures provided by the platform include (1) lipids, fatty acids and low-molecular-weight metabolites ($N = 51$); (2) lipid composition and particle concentration measures of lipoprotein subclasses ($N = 98$); (3) metabolite ratios ($N = 81$). This metabolomics platform has been extensively used in large-scaled epidemiological studies in the field of diabetes, cardiovascular disease, mortality and alcohol intake (*Akker et al., 2019*; *Würtz et al., 2016*; *Wurtz et al., 2012*; *Würtz et al., 2015*; *Fischer et al., 2014*). The data contained missing values due to detection limits. Samples with more than 25 missings were removed ($N = 71$), metabolites with more than 250 missings were removed ($N = 1$). Other missing values were replaced with the median value per metabolite. In total 231 metabolites in 2910 samples remained for analysis.

## Building biological clocks for multiple omics domains

Telomere length was multiplied by −1 to be able to compare directions of effects consistent with that of other biological clocks. For each of the other four *omics* domains (epigenetic, transcriptomic, metabolomic, and proteomic data) the same approach was used to compute biological clocks. First, the omics data were residualized with respect to technical covariates (batch, lab). Second, data per omics marker were normalized using a quantile-normal transformation. Finally, biological age was computed using cross-validation by splitting the sample in 10 equal parts. For each of the 10 groups, nine parts were used as training set and the 10th as test set. In the training set the biological age estimator was computed using ridge regression (R library glmnet), with chronological age as the outcome, and the omics data as predictors. Only for methylation and gene expression a selection of predictors (CpGs for methylation-based models and genes for gene-expression-based models) was made for each cross validation step: we increased the number of sites included in the elastic net in steps (steps for CpGs: 0, 100, 1000, 10,000, 80,000, 100,000, steps for gene expression 100, 500, 1000, 1200, 1400). CpGs/genes were selected in the order of their ranks derived from the association with age in the training sample. We selected the number of CpGs/genes where the cumulative

association signal reached a stable plateau. This approach is based on the rationale that adding more markers should theoretically never decrease predictive power. We previously performed tests where the number of CpGs/genes was included in the loop over the k-folds. However, as it produced very similar results but is much more computer intensive (*Clark et al., 2020*), this latter approach was not used. This approach resulted in 80,000 CpGs (mapping to 2976 genes) for the epigenetic clock, and 1200 probes (mapping to 767 genes) for the transcriptomic clock. For the proteomic and metabolomic data, all markers were used to predict age, because leaving markers out decreased the prediction accuracy. The predictor was then used in the test set to create an unbiased omics-based biological age. For each omics domain, biological aging was defined as the residuals of regressing biological age on chronological age (*Han et al., 2018*; *Peters et al., 2015*). Thus, in the terminology we use here, the biological aging indicators represent the biological age acceleration: a positive value means that the biological age is larger than the chronological age. A composite index of biological aging was made by scaling each of the five biological indicators and taking the sum, in the 653 samples that had data for all five omics levels.

## Health determinants

### Lifestyle

Alcohol consumption was assessed as units per week by using the AUDIT (*Babor et al., 1989*). Smoking status was assessed by pack years (smoking duration * cigarettes per day/20). Physical activity (*Gerrits et al., 2013*) was assessed using the International Physical Activity Questionnaire (IPAQ) (*Craig et al., 2003*) and expressed as overall energy expenditure in Metabolic Equivalent Total (MET) minutes per week (MET level * minutes of activity * events per week).

### Somatic health

BMI was calculated as measured weight divided by height-squared. Functional status is one of the most potent health status indicators in predicting adverse outcomes in aging populations (*Guralnik et al., 1996*), including depression (*Milaneschi and Penninx, 2014*). Assessment of functional status includes measures of physical impairments and disability, reflecting how individuals' limitations interact with the demands of the environment. Two measures of physical impairments were available: Lung capacity was determined by measuring the peak expiratory flow (PEF in liter/minute) using a mini Wright peak flow meter. Hand grip strength was measured with a Jamar hand held dynamometer in kilograms of force and was assessed for the dominant hand. Furthermore, physical disability was measured with the World Health Organization Disability Assessment Schedule II (WHODAS-II)s the sum of scale 2 (mobility) and scale 3 (self-care). The number of self-reported current somatic diseases for which participants received medical treatment was counted. We used *somatic disease categories* as categorized previously (*Gerrits et al., 2013*; *Gaspersz et al., 2018*): cardiometabolic, respiratory, musculoskeletal, digestive, neurological and endocrine diseases, and cancer. Metabolic syndrome components included waist circumference, systolic blood pressure, HDL cholesterol, triglycerides, and glucose levels, which measurement methods are described elsewhere (*Révész et al., 2014*).

### Mental health

Presence of current (6 month recency) major depressive disorder was assessed by the DSM-IV Composite International Diagnostic Interview (CIDI) version 2.1. Depressive severity levels in the week prior to assessment were measured with the 28-item Inventory of Depressive Symptomatology (IDS) self-report (*Rush et al., 1996*). Childhood trauma was assessed with the Childhood Trauma Interview (CTI) (*de Graaf et al., 2002*). In this interview, participants were asked whether they were emotionally neglected, psychologically abused, physically abused or sexually abused before the age of 16. The CTI reports the sum of the categories that were scored from 0 to 2 (0: never happened; 1: sometimes; 2: happened regularly), which was categorized into five categories.

## Statistical analyses

For each of the five biological aging indicators we computed associations with demographic (sex, education), lifestyle (physical activity, smoking, alcohol use), somatic health (BMI, hand grip strength, lung function, physical disability, chronic diseases), and mental health (current depression,

depression severity, childhood trauma) determinants using linear models with health determinants as predictors and biological aging as outcome (for each health determinant separately). All models included a covariate for sex, except for when sex was the outcome. For telomere length, chronological age was used as covariate in the models, for the other biological aging indicators age was not used as covariate because they are independent of chronological age by design. Standardized betas from these models are reported (by scaling predictor and outcome). Correction for multiple testing was done using permutation based FDR (*Fehrmann et al., 2011*). Subject labels were permuted 1000 times and associations were computed using the permuted data (all biological aging indicators vs all health determinants). For each of the observed p-values (*p*) the FDR was computed as the average number of permuted p-values smaller than *p*, divided by the amount of real p-values smaller than *p*, resulting in a p-value threshold of 2e-2 for a FDR of 5% for all tests. In the 653 overlapping samples with data in each biological clock domain, we scaled (mean 0, standard deviation 1) and summed up the five biological aging indicators in order to create a composite index of biological aging.

## Longitudinal analysis of mortality and chronic disease onset

As NESDA is a longitudinal study, with several follow-up measurement waves, we conducted post-hoc analyses on the relationship between the biological aging indicators and subsequent outcomes after six years of follow-up duration. The average chronological age of our cohort (mean = 41 years, sd = 13, range = 18–65 years) is rather young, so high rates of mortality and morbidity were not expected. Mortality data was gathered at each measurement wave. Also, at each wave self-reported somatic diseases for which participants received medical treatment were assessed. Based on this, we created *somatic disease categories* as categorized previously (*Gerrits et al., 2013*; *Gaspersz et al., 2018*): cardiometabolic, respiratory, musculoskeletal, digestive, neurological and endocrine diseases, and cancer. For these categories, we computed chronic disease onset defined as the disease not being present at baseline (time of biological aging assessment) and present at the latest wave (6 years after baseline). For each biological clock, we computed longitudinal analyses, using a linear model with mortality or chronic disease onset as outcome, and the biological clock residualized for chronological age as predictor, while correcting for sex.

# Acknowledgements

The infrastructure for the NESDA study (http://www.nesda.nl) is funded through the Geestkracht program of the Netherlands Organisation for Health Research and Development (ZonMw, grant number 10-000-1002) and financial contributions by participating universities and mental health care organizations (VU University Medical Center, GGZ inGeest, Leiden University Medical Center, Leiden University, GGZ Rivierduinen, University Medical Center Groningen, University of Groningen, Lentis, GGZ Friesland, GGZ Drenthe, Rob Giel Onderzoekscentrum). Telomere length assaying was supported through a NWO-VICI grant (number 91811602). Methylation sequencing was supported by NIMH grant R01MH099110. Metabolomics data were generated within the framework of the BBMRI Metabolomics Consortium funded by BBMRI-NL, a research infrastructure financed by the Dutch government (NWO, grant nr 184.021.007 and 184033111). Gene expression data were funded by the US National Institute of Mental Health (RC2MH089951).

# Additional information

## Competing interests

Brenda WJH Penninx: has received research funding (not related to the current paper) from Boehringer Ingelheim and Jansen Research. The other authors declare that no competing interests exist.

## Funding

No external funding was received for this work.

## Author contributions

Rick Jansen, Formal analysis, Visualization, Methodology, Writing - original draft, Writing - review and editing; Laura KM Han, Josine E Verhoeven, Yuri Milaneschi, Brenda WJH Penninx, Writing - original draft, Writing - review and editing; Karolina A Aberg, Edwin CGJ van den Oord, Writing - review and editing

## Author ORCIDs

Rick Jansen ⓘ https://orcid.org/0000-0002-3333-6737
Laura KM Han ⓘ https://orcid.org/0000-0001-9647-3723

## Ethics

Human subjects: The NESDA study was approved by the Ethical Review Boards of participating centers, and all participants signed informed consent. The population and methods of the NESDA study have been described in more detail elsewhere (Hillary et al., 2019).

## Decision letter and Author response

Decision letter https://doi.org/10.7554/eLife.59479.sa1
Author response https://doi.org/10.7554/eLife.59479.sa2

# Additional files

## Supplementary files

• Source data 1. Data used for *Figure 3*.

• Source data 2. Data used for *Figure 4*.

• Supplementary file 1. Associations between biological aging and medication use.

• Supplementary file 2. Associations between biological aging (individual indicators and composite index) and health determinants in 653 overlapping samples. For each biological aging indicator linear models were fit with the health determinant as predictor, while controlling for sex. The analysis was limited to the 653 samples with all five data layers available. Beta's and p-values from these models are presented here. * Beta for telomere length was multiplied by $-1$ to compare with other biological aging indicators. All measures are coded such that higher values indicate advanced biological aging. Bold indicates FDR < 5%.

• Supplementary file 3. Longitudinal analysis of biological aging and mortality and chronic disease onset.

• Transparent reporting form

## Data availability

According to European law (GDPR) data containing potentially identifying or sensitive patient information are restricted; our data involving clinical participants are not freely available in a public repository. However, we highly value scientific collaboration, therefore, in principle, NESDA data are available to all scientific researchers working at non-commercial research organizations worldwide. Researchers can request either existing data for data analyses or bioanalysis. Please visit the online data overview for an extensive overview of the available data and NESDA's current output (http://www.nesda.nl). Data are available upon request via the NESDA Data Access Committee (nesda@ggzingeest.nl). Gene expression data used for this study are available at dbGaP, accession number phs000486.v1.p1 (https://www.ncbi.nlm.nih.gov/projects/gap/cgi-bin/study.cgi?study_id=phs000486.v1.p1). As agreed with NIH and approved by the local IRB, the data was scheduled to be deposited in the NIH controlled access repository dbGAP. However, dbGAP is currently full and as soon as the new NIH controlled access repository comes online data will be deposited.

The following previously published dataset was used:

| Author(s) | Year | Dataset title | Dataset URL | Database and Identifier |
|---|---|---|---|---|
| Sullivan PF, Wright F, Wang W, Sun W, Zou F, Batista A, Madar V, Via K, Brooks A, Tischfield J, Wang Q, Qu A, Kochar J, D'Ambrosia D, Penninx B, Smit J, Jansen R, van Grootheest G, Boomsma D, deGeus E, Willemsen G, Hottenga J-J | 2014 | Integration of Genomics and Transcriptomics in unselected Twins and in Major Depression | https://www.ncbi.nlm.nih.gov/projects/gap/cgi-bin/study.cgi?study_id=phs000486.v1.p1 | dbGAP, phs000486.v1.p1 |

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
