## [Decision Letter]

**Acceptance summary:**

The paper by Jansen et al. investigates somatic and mental health using molecular substrates from the same individuals to create five different multi-omics biological aging clocks. This is exactly the kind of data needed to advance the field and to understand how different layers of data can be integrated to understand biological aging processes.

**Decision letter after peer review:**

Thank you for submitting your article "An integrative study of five biological clocks in somatic and mental health" for consideration by *eLife*. Your article has been reviewed by four peer reviewers, including Sara Hägg as the Reviewing Editor and Reviewer #1, and the evaluation has been overseen by Jessica Tyler as the Senior Editor. The following individuals involved in review of your submission have agreed to reveal their identity: Daniel W Belsky (Reviewer #2); Erik van den Akker (Reviewer #3); David G Le Couteur (Reviewer #4).

The reviewers have discussed the reviews with one another and the Reviewing Editor has drafted this decision to help you prepare a revised submission.

In this article, Jansen et al. take some steps toward addressing a knowledge gap on different molecular data used to derive biological aging clocks. They train algorithms to predict chronological age based on several molecular substrates assayed from blood samples: DNA methylation, gene expression, metabolomics, proteomics, and also telomere length. They then test correlations among the several derived measures and compare their associations with various exposures and criterion endpoints relevant to the aging process. The main finding is that the age-correlated features of the different substrates are (a) not very well correlated with one another and (b) largely non-overlapping in their information about health and exposure history. The authors also show that combining information across substrates produces a superior measurement as compared to substrate-specific measures.

1) Using established clocks

The authors talk about the problem of using their results in other studies since the omics platforms are not always available in other cohorts. It is not clear why it was necessary to train the algorithms in the NESDA cohort. There are published "clocks" for all of the substrates analyzed here (many examples are cited in the Introduction). For example, is there any way that the authors can calculate some of the standard epigenetic clocks (Horvath, Hannum, PhenoAge, GrimAge etc) perhaps not using the online calculator but using standalone scripts for these clocks adapted to your data format? Then, the results would be much more interesting for a wider audience and generalizable to other studies as well. The manuscript would be substantially stronger if these clocks were used in place of bespoke versions derived from the data used for testing hypotheses. If it is not feasible to implement published clocks in the NESDA data, this needs to be explained to the reader.

2) Comments on the algorithms used

If established clocks cannot be used, the alternative strategy of training and testing clocks within a single dataset needs to be presented as the alternative along with specific acknowledgement of the limitations of this approach. The Ridge-regression method used to train the clocks requires the assumption that patterning of molecular markers across the chronological age distribution in the sample reflects biological changes that occur with aging. That assumption has important limitations in any sample, for example see Nelson et al., 2020 on mortality selection or discussion of cohort effects in Belsky et al., 2015 or 2020 . But, depending on how age relates to sampling in the NESDA, there could be further challenges here.

Please also specify whether the feature selection for CpGs/genes was done on the whole dataset, prior to cross-validation, or within the cross-validation loop. If the first, this would lead to reporting overoptimistic performances (overtraining), if the latter, OK; please state so in the manuscript. Please also indicate what step size was used.

Would Mahalanobis distance be a better/more interesting way of analysing the data (eg Bello and Dumancas, Curr Aging Sci, 2017)?

A further consideration to be addressed if the NESDA data are to be used in training the "clocks": prediction of chronological age is only one criterion endpoint used to develop biological aging measures. Recent DNA methylation algorithms including the PhenoAge Clock and the GrimAge Clock were developed from analysis of physiology and mortality data along with chronological age. Some acknowledgement is needed that chronological age is only one of several potential criteria on which to train these measures.

Finally, algorithms trained by applying machine learning analysis to fit high-dimensional molecular data to chronological age variation are hypothesized to measure biological processes of aging. But this is a hypothesis, not a fact. Before we can interpret a "clock" as a measure of biological aging, we must establish that it changes with advancing age, forecasts disease, disability, and mortality, and indicates more advanced/delayed aging in individuals with exposure histories linked to shorter/longer healthy lifespan. The authors should be commended for undertaking some of this testing in NESDA, although using established clocks would be a better alternative. Hence, caution is warranted in interpretation of findings.

3) The NESDA cohort

More detail on NESDA is required to help the reader understand its appropriateness as a setting for comparing measures of aging. How was the sample selected? When were biological measurements collected and what was the extent of attrition from the baseline sample at those time points? In addition, it is not clear when the various exposure and health outcome measurements were collected relative to the biological measurements used to compose the clocks. For each participant, were all the analyses done in a blood sample taken at the same time, or were the different methods applied to bloods taken at different times? How long ago were the samples taken? Are there any storage effects that might influence analyses? A figure illustrating the timeline of data collection would greatly improve clarity of the analysis design.

4) The composite index

The composite index of the 5 clocks was a nice addition to the results. Ideally, clocks, including the composite, are scaled prior to associations with outcomes, as effect sizes are directly compared in the paper. Please indicate whether this has been done; if not, evaluations should be made on the basis of significance only, if so, great! Please state so in the Materials and methods.

Would the results have been similar if the 5 clocks were used combined in multivariate models instead? For example, what happens when all five clocks are put in the same model as predictors with BMI as health outcome? Will they all still be important in the association or are some redundant? Alternatively, it might be useful to compare the current operationalization of the multi-substrate composite to one derived from a factor analysis instead. This is information that is useful to understand the stability of the results but now somehow missed using only one composite index.

5) The Results

The biggest concern with the study is the generalizability given that the samples come from a clinical cohort with individuals suffering from depression and anxiety disorders. About 26% of the study participants were healthy controls and should then be representing a more general population. Moreover, age range is 20-65 years, so probably misses the age groups where biological changes of old age become dominant. Different participants were analysed for each biomarker, and there was a full dataset on only a fraction (approx. 1/3) of the participants that had individual tests done. Any data on ethnicities in NESDA? It is important to perform sensitivity analyses in selected groups of NESDA addressing these concerns in all associations and conclude if the effects are similar or changed in any important way. This also needs to be addressed in the Discussion section.

Another issue is the direction of effects, since these samples and associations are based on cross-sectional data, the authors correctly state that no conclusion can be made on the cause and consequence pathways. However, the biological clocks are used as outcomes in the linear regression models, why? For somatic health and chronic conditions, these are often treated as outcomes in the models using biological age as predictors. In Figure 3, intuitively this is interpreted as the health determinants are the outcomes.

If NESDA is a longitudinal cohort collected about 15 years ago, is there no other follow-up data on somatic and mental health that can be used then using biological age as predictor and health as outcome?

The authors should be somewhat more circumspect in interpreting the clocks they derive. A conclusion of the article is that biological aging proceeds differently across different molecular substrates. This takes the derived measures too literally. Instead, the finding is that the correlates of chronological age in different molecular substrates are not very well correlated with one another.

Given metabolic syndrome and depression are often medicated conditions, are there any data on medications, or any suggestion that medications might influence biomarkers?

6) Data availability

The statement on data availability is not good enough. Why can some gene expression data be released but not other data on these individuals? Data that are anonymized (the identifier key is thrown away) are not considered as sensitive data and it should hence be possible to release more data in this manner.

[Editors' note: further revisions were suggested prior to acceptance, as described below.]

Thank you for resubmitting your work entitled "An integrative study of five biological clocks in somatic and mental health" for further consideration by *eLife*. Your revised article has been evaluated by Jessica Tyler (Senior Editor) and Sara Hägg (Reviewing Editor).

The current version of the manuscript represents a highly responsive revision that addressed most comments. There are some remaining issues that need to be addressed before acceptance, as outlined below:

– The mortality association should be mentioned already in the result section as an additional analysis.

– The biological aging indicator is not clearly described in all figure legends, if it is the residulized age version or not.

– State the direction of effect for the medication analysis in the result section, not just p-values.

– The metabolomic platform is "Brainshake" in Materials and methods?

---

## [Author Response]

Essential revisions:1) Using established clocksThe authors talk about the problem of using their results in other studies since the omics platforms are not always available in other cohorts. It is not clear why it was necessary to train the algorithms in the NESDA cohort. There are published "clocks" for all of the substrates analyzed here (many examples are cited in the Introduction). For example, is there any way that the authors can calculate some of the standard epigenetic clocks (Horvath, Hannum, PhenoAge, GrimAge etc) perhaps not using the online calculator but using standalone scripts for these clocks adapted to your data format? Then, the results would be much more interesting for a wider audience and generalizable to other studies as well. The manuscript would be substantially stronger if these clocks were used in place of bespoke versions derived from the data used for testing hypotheses. If it is not feasible to implement published clocks in the NESDA data, this needs to be explained to the reader.

We thank the reviewers for this comment and explain below why we do not use standard epigenetic clocks (or other published clocks using the omics levels presented here), which, if possible, would indeed have facilitated interpretation and replication of our results.

It is important to note that commonly used methods for assaying DNA methylation depend on the Illumina arrays, platforms that generate variables indicating the percentage of methylated CpGs (values range from 0 to 1, indicating no methylation to 100% methylation), whereas the current study used optimized Methyl-CpG binding domain sequencing, that generate quantitative scores (e.g. scores ranging from 0-20) representing the number of fragments covering a CpG that is proportional to the level of methylation occurring in its locus. Optimized MBD-seq has a more comprehensive coverage of the methylome (interrogation of 94% instead of 2-4% of all 28 million common CpG sites in blood). Computing existing array-based epigenetic clocks (e.g. Horvath) using the MBD-seq data would be suboptimal compared to computing MBD-seq based clocks because: (1) MBD-seq covers much more CpGs than the array-based methylation, and restricting to array based CpGs means not leveraging that coverage, and (2) using only the CpGs used for the Horvath clock from the MBD-seq method does not generate equivalent predictive power.

As for the other omics-based clocks that we computed: there are existing gene expression based clocks (Peters et al., 2015) however, gene expression measures are also platform dependent. The mentioned existing clock used Illumina arrays, while we generated gene expression data from Affymetrix arrays, which (in many cases) target different parts of the gene and not completely overlapping gene sets. The proteomic clock is based on a selected number of proteins from a particular platform with 172 proteins. To the best of our knowledge, this platform has not been used before to compute a proteomic clock. The metabolomic clock was created using the Nightingale platform: this platform has been used before to compute biological age, and an algorithm to compute biological age was provided (van den Akker et al., 2019). However, NESDA was part of this study (and thus part of the “training set” to compute the algorithm): using this algorithm in NESDA may lead to overfitting. More importantly, the above mentioned transcriptomic and metabolomic clocks have not been validated by other independent studies. For the above outlined reasons, we believe that the previously established clocks are suboptimal and/or not suitable for the current available data. While we agree that this comes at the cost of generalizability of findings, we also ask for the reviewers understanding that applying existing clocks is not optimal with the available data. Importantly, by using one consistent approach for calculating the biological clocks using different omics data, we increase the comparability of clocks within our own study.

We now clarify this and write the following in the Discussion to emphasize the limitations of this strategy:

“This study did not include existing biological clocks. […] However, we emphasize the need for epidemiological replication of these determinants in other datasets (e.g. those including different ethnicities) and we recognize that data harmonization and pooling are important strategies on the scientific research agenda that may overcome this limitation in the future.”

2) Comments on the algorithms usedIf established clocks cannot be used, the alternative strategy of training and testing clocks within a single dataset needs to be presented as the alternative along with specific acknowledgement of the limitations of this approach. The Ridge-regression method used to train the clocks requires the assumption that patterning of molecular markers across the chronological age distribution in the sample reflects biological changes that occur with aging. That assumption has important limitations in any sample, for example see Nelson et al., 2020 on mortality selection or discussion of cohort effects in Belsky et al., 2015 or 2020 . But, depending on how age relates to sampling in the NESDA, there could be further challenges here.

We thank the reviewers for the notion of these references, and now write in the Discussion:

“Since no previously published algorithms were used, we trained our own clocks using ridge regression with cross-validation. […] Future longitudinal studies are needed to identify patterns of biological changes that go beyond their ability to predict age at the time of sampling.”

Please also specify whether the feature selection for CpGs/genes was done on the whole dataset, prior to cross-validation, or within the cross-validation loop. If the first, this would lead to reporting overoptimistic performances (overtraining), if the latter, OK; please state so in the manuscript. Please also indicate what step size was used.

We now clarified this in the Materials and methods section:

“We increased the number of sites included in the elastic net in steps (steps for CpGs: 0, 100, 1000, 10 000, 80 000, 100 000, steps for gene expression 100, 500, 1000, 1200, 1400). […] This approach resulted in 80,000 CpGs (mapping to 2,976 genes) for the epigenetic clock, and 1,200 probes (mapping to 767 genes) for the transcriptomic clock.”

Would Mahalanobis distance be a better/more interesting way of analysing the data (eg Bello and Dumancas, Curr Aging Sci, 2017)?

We thank the reviewer for this suggestion, but in order to make our results comparable to the most commonly previously used methods with similar research designs, we choose not to use the Mahalanobis distance.

A further consideration to be addressed if the NESDA data are to be used in training the "clocks": prediction of chronological age is only one criterion endpoint used to develop biological aging measures. Recent DNA methylation algorithms including the PhenoAge Clock and the GrimAge Clock were developed from analysis of physiology and mortality data along with chronological age. Some acknowledgement is needed that chronological age is only one of several potential criteria on which to train these measures.

We now acknowledge this in the Discussion:

“While the current study only used chronological age as criterion endpoint, it is important to mention that other epigenetic clocks exist that are trained to predict other potential criteria such as phenotypic markers of age (DNAm PhenoAge)^43^ or a composite biomarker that was derived from DNAm surrogates and smoking in pack-years (GrimAge)^44^. Such clocks were developed to lead to improved predictions of risk of mortality.”

Finally, algorithms trained by applying machine learning analysis to fit high-dimensional molecular data to chronological age variation are hypothesized to measure biological processes of aging. But this is a hypothesis, not a fact. Before we can interpret a "clock" as a measure of biological aging, we must establish that it changes with advancing age, forecasts disease, disability, and mortality, and indicates more advanced/delayed aging in individuals with exposure histories linked to shorter/longer healthy lifespan. The authors should be commended for undertaking some of this testing in NESDA, although using established clocks would be a better alternative. Hence, caution is warranted in interpretation of findings.

The reviewer is correct on these points, it is also our understanding that we cannot conclusively state that the biological clocks considered in the current cross-sectional study in fact reflect biological processes of ongoing aging. We agree that this limitation did not receive enough attention in the previous version of the manuscript. We have to acknowledge that our sample is rather young (average age=41, range 18-65 years). Despite the fact that we have longitudinal data on our respondents, it is to be expected that the power to look at aging-related outcomes such as mortality and disease onset in our sample is still limited. However, we now performed additional analyses in which we associated biological aging with mortality and chronic disease onset outcomes. These analyses showed no significant associations with biological aging indicators. We added these findings to the Results section and revised the text to reflect that caution is warranted in interpretation of our findings.

We now write in the Materials and methods section:

“Longitudinal analysis of mortality and chronic disease onset

As NESDA is a longitudinal study, with several follow-up measurement waves, we conducted post-hoc analyses on the relationship between the biological aging indicators and subsequent outcomes after six years of follow-up duration. […] For each biological clock we computed longitudinal analyses, using a linear model with mortality or chronic disease onset as outcome, and the biological clock residualized for chronological age as predictor, while correcting for sex.”

And further in the Discussion:

“Another aspect that limits the interpretability of our findings in the context of increased risk of developing aging-related diseases and mortality was the relatively young age of the current sample. […] Before definitively interpreting a "clock" as a measure of biological aging, further independent studies are needed to establish that the clock changes with advancing age and forecasts disease, disability and mortality.”

3) The NESDA cohortMore detail on NESDA is required to help the reader understand its appropriateness as a setting for comparing measures of aging. How was the sample selected? When were biological measurements collected and what was the extent of attrition from the baseline sample at those time points? In addition, it is not clear when the various exposure and health outcome measurements were collected relative to the biological measurements used to compose the clocks. For each participant, were all the analyses done in a blood sample taken at the same time, or were the different methods applied to bloods taken at different times? How long ago were the samples taken? Are there any storage effects that might influence analyses? A figure illustrating the timeline of data collection would greatly improve clarity of the analysis design.

We thank the reviewer for their comment to add more detailed information of NESDA in the manuscript. We now point out in the revised description that all health determinants and blood-based biological clocks were collected at the same timepoint, making a figure illustrating the timeline of data collection seem redundant as there is also no attrition to report. However, we now improved the clarity of the analysis design by adding a section on the timing of data collection, storage, and detailed report of the baseline interview. The Materials and methods section now reads:

“Data used were from the Netherlands Study of Depression and Anxiety (NESDA), an ongoing longitudinal cohort study examining course and consequences of depressive and anxiety disorders. […] Data to derive different biological clocks was available for different subsamples and all based on the same fasting blood draw from participants in the morning between 8:30 and 9:30 after which samples were stored in a -80°C freezer or – for RNA – transferred into PAXgene tubes (Qiagen, Valencia, California, USA) and stored at −20°C. To create biological clocks, we used telomere length (*N*=2936), DNA methylation (*N*=1130, MBD-seq, 28M CpGs), gene expression (*N*=1990, Affymetrix U219 micro arrays, >20K genes), proteins (*N*=1837, Myriad RBM DiscoveryMAP 250+, 171 proteins) and metabolites (*N*=2910, Brainshake platform, 231 metabolites), see Table 1 and details in the following sections.”

4) The composite indexThe composite index of the 5 clocks was a nice addition to the results. Ideally, clocks, including the composite, are scaled prior to associations with outcomes, as effect sizes are directly compared in the paper. Please indicate whether this has been done; if not, evaluations should be made on the basis of significance only, if so, great! Please state so in the Materials and methods.

We indeed standardized all our clocks prior to association analyses. So, the reported associations are standardized betas that can be compared across individual biological aging indicators. We better indicated this now in the Materials and methods:

“Standardized betas from these models are reported (by scaling predictor and outcome).”

And

“A composite index of biological aging was made by scaling each of the five biological indicators and taking the sum, in the 653 samples that had data for all five omics levels.”

Would the results have been similar if the 5 clocks were used combined in multivariate models instead? For example, what happens when all five clocks are put in the same model as predictors with BMI as health outcome? Will they all still be important in the association or are some redundant? Alternatively, it might be useful to compare the current operationalization of the multi-substrate composite to one derived from a factor analysis instead. This is information that is useful to understand the stability of the results but now somehow missed using only one composite index.

We agree that there are multiple strategies to compute a composite index. As an alternative we now also report the associations between the first PC of a PCA analysis of the 5 biological aging indicators, and the health determinants. This PC reflects the strongest biological aging correlations, which is between metabolomic and proteomic aging. Thus, this PC is a weighted sum of the indicators, all positive weights, but with highest weights for metabolomic and proteomic aging indicators. Therefore, the associations between this PC and the health outcomes, is very similar to the associations between the health determinants and metabolomic and proteomic aging. Moreover, PCA analysis gives the explained variance of multiple independent dimensions derived from the underlying data, and therefore also helps to answer the reviewers question about redundancy of the biological clocks. We added to the Results:

“As an alternative approach, Principal Component Analysis (PCA) was used to compute an alternative composite index. […] The five PC’s each explain more than 15% of variance (the first 2 PC’s more than 25% each), indicating the multidimensionality and non-redundancy of the five biological clocks.”

5) The ResultsThe biggest concern with the study is the generalizability given that the samples come from a clinical cohort with individuals suffering from depression and anxiety disorders. About 26% of the study participants were healthy controls and should then be representing a more general population. Moreover, age range is 20-65 years, so probably misses the age groups where biological changes of old age become dominant. Different participants were analysed for each biomarker, and there was a full dataset on only a fraction (approx. 1/3) of the participants that had individual tests done. Any data on ethnicities in NESDA? It is important to perform sensitivity analyses in selected groups of NESDA addressing these concerns in all associations and conclude if the effects are similar or changed in any important way. This also needs to be addressed in the Discussion section.

We would like to emphasize that sample size is important for testing robust associations, so we wanted to make optimal use of the current sample sizes to examine individual health determinant associations. The rationale to additionally test all associations in a fully overlapping sample with all omics data (N=653) was to compare whether shared and unique associations would be (much) different, but this analysis showed very similar results to those with a maximum number of samples per omics level, indicating high sensitivity of the results.

Furthermore, although we believe that poorer mental or somatic health outcomes will be similarly associated to advanced aging in both control and patient groups, we can see how associations might be confounded by diagnostic status. We therefore repeated the same separate univariate analyses, but included an additional interaction term between current depression status and health determinant in the models. Overall, these interaction terms were not significant. These findings can be found in the supplement.

We now write in the Discussion:

“To further examine whether the results were consistent across participants with and without depressive psychopathology, we repeated all models in post-hoc analyses and added an interaction term between current depression status and health determinants. There was an overall consistent pattern of non-significant interaction terms for most health determinants and biological aging, although only higher BMI was significantly associated to advanced epigenetic aging in the psychopathology group. However, taken together, the results suggest that findings are not different in persons with and without mental disorders.”

We have also added the following sentence to the Materials and methods:

“More than 94% of the NESDA participants were from North European origin.”

And the following sentence in the Discussion:

“We also emphasize the need for epidemiological replication of these determinants in other datasets (e.g. those including different ethnicities) and we recognize that data harmonization and pooling are important strategies on the scientific research agenda that may overcome this limitation in the future.”

Another issue is the direction of effects, since these samples and associations are based on cross-sectional data, the authors correctly state that no conclusion can be made on the cause and consequence pathways. However, the biological clocks are used as outcomes in the linear regression models, why? For somatic health and chronic conditions, these are often treated as outcomes in the models using biological age as predictors. In Figure 3, intuitively this is interpreted as the health determinants are the outcomes.

In cross sectional analysis, interchanging predictor and outcome will not significantly change the interpretation of associations between these variables in a linear model. Since we wanted to find determinants for biological aging, while correcting for sex, we used biological aging as outcomes. We now clarify in the legend of Figure 3 that biological clocks residualized for age were used as outcomes.

If NESDA is a longitudinal cohort collected about 15 years ago, is there no other follow-up data on somatic and mental health that can be used then using biological age as predictor and health as outcome?

We appreciate this comment and have conducted additional analyses to examine whether biological aging at baseline predicts the presence of somatic disease and mortality rates 6 years later in those who were initially disease-free at baseline. Please see response 2.

The authors should be somewhat more circumspect in interpreting the clocks they derive. A conclusion of the article is that biological aging proceeds differently across different molecular substrates. This takes the derived measures too literally. Instead, the finding is that the correlates of chronological age in different molecular substrates are not very well correlated with one another.

We assume the reviewers refer to this sentence in the Discussion: “Biological aging seems to be differently manifested at certain cellular levels, as suggested by the range of correlations among the biological clocks considered in this study.”

To make clearer that findings from the biological clocks are only suggesting findings for biological aging we changed this sentence into:

“The range of correlations among the biological aging indicators considered in this study indicates that the correlates of chronological age in different molecular layers were not strongly correlated, suggesting that biological aging may be differently manifested at certain cellular levels.”

Given metabolic syndrome and depression are often medicated conditions, are there any data on medications, or any suggestion that medications might influence biomarkers?

We thank the reviewer for this suggestion, and provide additional analyses in which we verified if antidepressant medication (SSRIs, TCAs, or other antidepressants) or metabolic syndrome related medication (anti-diabetic, fibrates, or anti-hypertensives), were associated with the biological clocks.

The findings can be read in the Results section:

“To verify if the results were confounded by medication use, we computed associations between antidepressant medication (SSRIs, TCAs, or other antidepressants), metabolic syndrome related medication (“metabolic medication”: anti-diabetic, fibrates, or anti-hypertensives) and biological aging (Supplementary file 1). After FDR correction, we found that metabolomic aging was associated with the use of metabolic medication (*P*=2.35e-3), and antidepressant use with proteomic (*P*=7.16e-5) and transcriptomic aging (*P*=8.1e-3). The design of the current observational study cannot conclusively prove whether this is a direct medication effect or confounding by indication.”

We believe, however, that these findings are not indicating direct medication effects per se. In fact, these could be interpreted as confirmation of our disease findings. E.g. the patient group using antidepressant medication is also the group that is more chronically and severely depressed. Therefore, using the current data from an observational cohort, we are unable to discern the effects of depression, depression severity, or direct effects of antidepressants. The same is true for the findings of metabolic medication.

We have added the following sentence to the Discussion:

“We observed some significant associations between biological aging and medication use. The design of the current observational study cannot conclusively prove whether this is a direct medication effect or confounding by indication: the patient group using antidepressant medication is also the group that is more chronically and severely depressed. This is similar for the metabolic syndrome related medication. Future studies using randomized clinical trial designs are needed to investigate the mechanism of action of direct pharmacological effects of medication on biological aging.”

6) Data availabilityThe statement on data availability is not good enough. Why can some gene expression data be released but not other data on these individuals? Data that are anonymized (the identifier key is thrown away) are not considered as sensitive data and it should hence be possible to release more data in this manner.

We have updated our data availability section.

[Editors' note: further revisions were suggested prior to acceptance, as described below.]

The current version of the manuscript represents a highly responsive revision that addressed most comments. There are some remaining issues that need to be addressed before acceptance, as outlined below:– The mortality association should be mentioned already in the result section as an additional analysis.

We now write in the Results section:

“Association between biological aging indicators and mortality in longitudinal analysis

We conducted post-hoc analyses on the relationship between the biological aging indicators and subsequent outcomes after six years of follow-up duration. Mortality data and self-reported somatic disease onset (in the categories cardiometabolic, respiratory, musculoskeletal, digestive, neurological and endocrine diseases, and cancer) was gathered at each measurement wave. There were no significant associations between chronic disease onset or mortality and baseline biological aging, likely due to the low numbers of mortality and disease onset (Supplementary file 3).”

– The biological aging indicator is not clearly described in all figure legends, if it is the residulized age version or not.

We added to the legends of Figure 3 and Figure 4:

“All biological aging indicators were age-regressed, only telomere length was not.”

– State the direction of effect for the medication analysis in the result section, not just p-values.

We now write in the Results section:

“After FDR correction, we found that metabolomic aging was associated with the increased use of metabolic medication (*Β=0.153, P=2.35*e-3), and antidepressant use with proteomic (*Β=0.208, P=*7.16e-5) and transcriptomic aging (*Β=0.129, P=8.1e-3*). The design of the current observational study cannot conclusively prove whether this is a direct medication effect or confounding by indication.”

– The metabolomic platform is "Brainshake" in Materials and methods?

We corrected this to “Nightingale platform” (formerly known as “Brainshake”).